# Reshaping Our Knowledge: Advancements in Understanding the Immune Response to Human Respiratory Syncytial Virus

**DOI:** 10.3390/pathogens12091118

**Published:** 2023-09-01

**Authors:** Federica Attaianese, Sara Guiducci, Sandra Trapani, Federica Barbati, Lorenzo Lodi, Giuseppe Indolfi, Chiara Azzari, Silvia Ricci

**Affiliations:** 1Postgraduate School of Pediatrics, University of Florence, Meyer Children’s Hospital IRCCS, 50139 Florence, Italy; federica.barbati@unifi.it; 2Postgraduate School of Immunology, University of Florence, Meyer Children’s Hospital IRCCS, 50139 Florence, Italy; sara.guiducci@unifi.it; 3Pediatric Unit, Meyer Children’s Hospital IRCCS, Viale Pieraccini 24, 50139 Florence, Italy; sandra.trapani@unifi.it (S.T.); giuseppe.indolfi@unifi.it (G.I.); 4Department of Health Sciences, University of Florence, 50139 Florence, Italy; lorenzo.lodi@unifi.it (L.L.); chiara.azzari@unifi.it (C.A.); 5Division of Immunology, Meyer Children’s Hospital IRCCS, Viale Pieraccini 24, 50139 Florence, Italy; 6NEUROFARBA Department, University of Florence, 50139 Florence, Italy

**Keywords:** immune response, RSV, syncytial respiratory virus, prevention strategy

## Abstract

Human respiratory syncytial virus (hRSV) is a significant cause of respiratory tract infections, particularly in young children and older adults. In this review, we aimed to comprehensively summarize what is known about the immune response to hRSV infection. We described the innate and adaptive immune components involved, including the recognition of RSV, the inflammatory response, the role of natural killer (NK) cells, antigen presentation, T cell response, and antibody production. Understanding the complex immune response to hRSV infection is crucial for developing effective interventions against this significant respiratory pathogen. Further investigations into the immune memory generated by hRSV infection and the development of strategies to enhance immune responses may hold promise for the prevention and management of hRSV-associated diseases.

## 1. Introduction

Human respiratory syncytial virus (hRSV) is a leading cause of acute lower respiratory tract infections (LRTIs), such as pneumonia and bronchiolitis, in infants and young children worldwide. It is the most frequent cause of hospitalization in children under two years of age, and it is associated with an increased risk of developing recurrent wheezing and asthma [1,2]. After pneumococcal pneumonia, RSV is also a significant cause of death in childhood from LRTI [3,4]. It is second only to malaria as a cause of death in children between 1 month and one year of age [5]. Most hRSV infections occur in previously healthy children, but children <3 months of age and with particular risk factors, such as preterm birth, chronic pulmonary disease, congenital heart disease, neuromuscular disorders, and immunodeficiencies are at high risk of severe evolution of RSV infections [6].

Furthermore, hRSV infection is a significant cause of morbidity in adults with comorbidity and elders [2,7,8,9]. Currently, there is no effective treatment available for hRSV disease but several prevention strategies, such as vaccines and monoclonal antibodies (mAbs), are under investigation. To date, immune prophylaxis is limited to Palivizumab, which is an anti-RSV fusion (F) protein monoclonal antibody. Palivizumab is approved for use in children who were born before 36 weeks of gestation and are younger than six months old at the start of the RSV season. It is also recommended for children under two years of age who have been treated for bronchopulmonary dysplasia (BPD) within the last six months or have hemodynamically significant congenital heart disease [10]. However, it is important to note that regional guidelines may differ. Recently, a new extended half-life mAb to the RSV fusion protein (Nirsevimab) was approved by the European Medicines Agency (EMA) [11]. A single dose of Nirsevimab has been shown to protect healthy infants born at term and preterm from medically attended hRSV LRTIs, associated hospital admission, and severe hRSV infection [12,13]. This narrative review aims to provide an update on the immune response to hRSV infection, underpinning new prevention and treatment strategies.

## 2. Human Respiratory Syncytial Virus’ Generalities

### 2.1. Molecular Structure and Function

HRSV belongs to the Mononegavirales order, Paramyxoviridae family, and Pneumovirus genus. Two major serotypes are currently known: hRSV A and hRSV B. The hRSV virions consist of a nucleocapsid within a lipid envelope of irregular spherical shape sized from 150 to 300 nm (Figure 1). The HRSV genome consists of an unsegmented, single-stranded, negative polarity linear RNA of approximately 15.2 kb organized in ten genes encoding 11 proteins (Table 1): two functional proteins, NS1 and NS2, the nucleocapsid proteins N, the polymerase cofactor phosphoprotein P, the matrix protein M, the hydrophobic viroporin SH, the glycoprotein G, the fusion protein F, the matrix protein M2-1 and M2-2, and the RNA polymerase-dependent RNA nucleocapsid L protein [2]. The viral genome is enclosed within a protein-based nucleocapsid composed of N, P, L, M2-1, and M2-2 proteins.

The nucleoprotein N is the main capsid component in close contact with the viral genome. It is thought to protect the viral RNA from nucleases and, together with P and L proteins, it constitutes the RSV ribonucleoprotein (RNP) complex, which regulates the transcription and replication of the viral RNA [14,15]. As the principal role of the L protein, the RNA-dependent RNA polymerase is the replication and transcription of the viral genome, which is supported by the RNP complex. As hRSV is a negative-sense RNA virus, the L protein directly transcripts the genome into mRNA for each hRSV gene’s expression, recognizing the promoter region in the leading sequence at the 3′ viral RNA strand extreme [2,16]. The M2 gene codes M2-1 and M2-2 through two open-reading frames (ORFs). These proteins are involved in the transcriptional and replication process. In detail, M2-1 cooperates with the L protein as an elongation and anti-terminator factor during the transcriptional phase while, in the replicative phase, it participates in the assembly of the viral particle by acting as a bridge between the M protein of the matrix and the RNP complex. The M2-2 protein, an inhibiting viral transcription, is directly involved in maintaining the balance between viral genome replication and transcription [17,18].

The matrix protein M is a non-glycosylated, phosphorylated protein of about 256 amino acids, which has an important role in the virions’ assembly and replication. Beyond its structural function, the M protein can inhibit viral transcription and interact with hRSV G and F proteins to signal the assembly of the virions. Moreover, in the nucleus, the M protein can reduce the transcription activity of the host cell and arrest the cell cycle in the G1 phase. These actions, which are p53-dependent, increase hRSV replication [2,29,30].

The hRSV viral envelope consists of a double phospholipid layer combined with three membrane proteins: glycoprotein G, fusion glycoprotein F, and the small hydrophobic protein SH (Figure 1). Among these, G and F proteins play a pivotal role in the infectivity and pathogenesis of hRSV. The G protein mediates the adherence of the virus to the host cell, while the F protein allows the fusion of the viral envelope with the host membrane cell by changing its conformation to a post-fusion form. These proteins are also directly responsible for stimulating a targeted antibody response in the host [33].

The antigenic variability between hRSV’s two main serotypes, RSV A and RSV B, is mainly determined by the protein G, which provides only 35% of homology between RSV A and B. In detail, protein G is a glycoprotein synthesized as a polypeptide composed of about 300 amino acids with a hydrophobic domain at the C-terminal, which is characterized by a strong immunogenic component [19,20]. In this domain, three types of different antigenic epitopes have been identified: (I) conserved epitopes, present in all viral strains, (II) group-specific epitopes, expressed only by the same antigenic group, and (III) strain-specific epitopes, present only in the specific strain of the same antigenic group expressed in the hyper-variable region C-terminal of the hydrophobic domain of protein G [21].

Protein F is a transmembrane surface glycoprotein highly conserved between hRSV serogroups with less than 10% of sequence diversity between the A and B groups [2]. It is synthesized as an inactive precursor of 67 kD (574 amino acids) indicated as F0 in the host cell’s cytoplasm. It is subsequently converted into an active protein following proteolytic cleavage by a protease within the Golgi apparatus. The proteolytic cleavage generates three hydrophobic polypeptides: (I) the signal peptide at the N-terminal end, which mediates the translocation of the new polypeptide within the lumen of the endoplasmic reticulum; (II) a transmembrane region, close to the C-terminal end, which binds the F protein to the cell and viral membranes; and (III) the fusion peptide which determines the fusion process. Once the F fusion pre-protein has bound to the cell surface, it changes conformation, becoming active. At this point, the activated protein F promotes the fusion of the viral membrane with the host cell membrane with the consequent transfer of the viral genetic material [22]. In addition, the activated protein F promotes the fusion of infected cells of the respiratory epithelium adjacent to each other, thus promoting the formation of the typical *syncytia*. Notably, the F protein can interact with different proteins on the surface of host cells, such as TLR4, 37 ICAM-1, and nucleolin [2,23].

The SH protein is found on the surface membrane of the virion and can exist in two different forms of varying sizes, depending on the hRSV serotype [24]. It belongs to the family of small hydrophobic viral proteins and is involved in the hRSV pathogenetic process. It forms a pentameric ionic channel, increasing the membrane’s permeability in the host cell. Deletion of the SH protein in RSV induces viral attenuation. Furthermore, the role of the SH protein in the anti-apoptotic process in infected cells was recently found [2,25,26,27,28].

The non-structural proteins NS1 and NS2 prevent the host cell from producing type 1 interferons (IFN-I) and promote phosphoinositide 3-kinase (PI3K) pathways inhibiting apoptosis, promoting the survival of infected cells, and increasing viral yield [31,32,34].

### 2.2. HRSV Entry into the Host Cell

The main target cell types for HRSV are ciliated cells in the airway epithelium and type-1 alveolar pneumocytes [35]. HRSV entry into the host cell is ensured by the interaction between the cellular membrane receptors and the viral envelope proteins G and F [33,36] (Figure 1). Many membrane proteins have been identified as viral receptors: annexin II, chemokine CX3 receptor 1 (CX3CR1), epidermal growth factor receptor (EGFR), Toll-like receptor 4 (TLR-4), intercellular adhesion molecule 1 (ICAM-1), nucleolin protein, heparan sulfate membrane proteoglycan receptors (HSPGs), and insulin growth factor 1 receptor (IGF1R) (Table 2) [36,37]. CX3CR1, the nucleolin protein, and TLR-4 are the most immunologically relevant of the latter. CX3CR1 is widely expressed on the ciliated cell membrane of the bronchial epithelium, which is the leading hRSV replication site. The G protein interacts with CX3CR1 through a CX3C domain (Figure 1) whose mutation may reduce the pathogenicity of the virus. In CX3CR1-defective mice, a decreased susceptibility to hRSV infection was found [38]. In addition, preclinical studies demonstrated that the RSV/CX3CR1 interaction induces CCL5, IL-8, and fractalkine production whilst downregulating IL-15, IL1-RA, and monocyte chemotactic protein-1. Thus, the RSV-G/CX3CR1 interaction is likely crucial in infection and infection-induced responses of the airway epithelium [38,39,40]. Toll-like 4 receptor (TLR-4) is another surface protein the ciliated cells express. The interaction between TLR-4 and the F protein (RSV-F) can induce protein kinases signaling cascade that may activate the endocytosis pathway [41] (Figure 1). Nucleolin is a ubiquitous molecule with many reported functions, including ligand binding and cargo shuttling between the cell surface, cytosol, and nucleus, which is more expressed by actively proliferating cells such as alveolar cells in children up to 2 years of age. In vitro studies have shown that the human nucleolin protein’s induced expression in insect cells can make them susceptible to hRSV infection [23,42]. Although the mechanism of fusion of the viral envelope with the cell membrane of the host is not yet fully understood, some studies have hypothesized that the pre-fusion RSV-F (RSV-F1) is a trimer characterized by antigenic epitopes in the N-terminal extremity protruding outside the viral envelope. The interaction RSV-F1/nucleolin induces a conformational change of the trimer and the release of the N-terminal fusion peptide, which inserts in the host cell membrane. This mechanism results in close contact between the virus envelope and host cell membrane, generating a fusion pore and the release of the viral nucleocapsid into the cell cytoplasm [43] (Figure 1).

## 3. Epidemiology

The hRSV can cause various respiratory tract diseases, ranging from mild upper respiratory tract infections to severe lower respiratory tract infections that may require hospitalization. The clinical syndrome of bronchiolitis is the most common severe disease manifestation [44]. Most children become infected with hRSV during the first year of life, and virtually all are infected by two years of age. Most hRSV LRTIs occur in healthy full-term infants during the first year of life, with children <6 months of age at a higher risk of death [44]. It is a common occurrence for individuals of all ages to experience multiple infections. However, it is important to note that prior infection does not necessarily guarantee immunity from future infections even if they occur in consecutive years [45,46,47]. In Europe, an average of 245,244 hospital admissions with respiratory infection associated with RSV per year was estimated in children under the age of 5 years, with most cases occurring among children aged less than 1 year (74.9%) [48]. Otherwise, 99% of RSV deaths occur in low-income countries without proper primary and supportive care access [3]. Risk factors for more severe hRSV disease can be divided into host, environmental, and viral factors [28,44]. The latest epidemiological data on the global burden of RSV infection in children and adults are summarized in Figure 2.

### 3.1. Host and Environmental Factors

It is increasingly apparent that certain genetic associations are correlated with susceptibility and the severity of respiratory syncytial virus (RSV). For instance, studies have shown that identical twins are often concordant for severe RSV disease. Indeed, some genetic polymorphisms of both innate and adaptative immunity genes like surfactant proteins, interleukins (IL-4, IL-8, IFN-I, RANTES), interleukin receptors (CCR5), host cell receptors (TLR2, TLR3, TLR4, RIG-I) and transcriptional factors (NF-κB) may contribute to determining the susceptibility and severity of hRSV infection [28,49,50].

Environmental factors that increase hRSV infection risk include tobacco smoke exposure, lack of breastfeeding, and low socioeconomic status [28,51].

#### 3.1.1. Infants

The impact of RSV is significantly greater as it can infect infants at an extremely young age. Very young infants are less tolerant of respiratory infections than older individuals, partly because they have narrower airways and, thus, are more susceptible to obstruction, which is a prominent feature of RSV disease [28]. Secondly, young infants develop a lower magnitude and less effective immune response, which seems related to immunologic immaturity immunosuppression by RSV-specific maternal serum antibodies [28,52,53]. Low birth weight and prematurity, cardiopulmonary chronic lung disease, neurodevelopmental disorders, and congenital or acquired immunodeficiency are significant risk factors for hospitalization.

#### 3.1.2. People Living with Primary or Secondary Immunodeficiency

Immunodeficiencies predispose to RSV infection and disease both in children and adults. People living with HIV/AIDS (PLWHA) are at increased risk of developing severe LRTI when infected by hRSV. Among them, children are 3.5 times more likely to be hospitalized and have a higher mortality rate than HIV-uninfected children [54,55]. HRSV is a common cause of LRTI in hematopoietic stem cell transplant recipients with an infection rate as high as 17% in some settings. Among this population, RSV pneumonia has up to 83% mortality. Several factors have been identified as linked to a higher severity of symptoms and poorer outcomes for hematopoietic stem cell transplant recipients becoming infected with RSV: male sex, older age, allogeneic graft, unrelated donor, myeloablative regimen, cytomegalovirus seropositivity, graft-versus-host disease (GVHD), and hRSV infection within the first 3 months following transplantation. [56,57].

#### 3.1.3. Elders

Many different factors seem to be involved in the increased susceptibility of the elderly to respiratory virus infections including RSV, such as the presence of comorbidities that can be rekindled by infection and the atypical clinical presentation that can lead to diagnostic delay. Moreover, two phenomena have been studied in recent years: immunosenescence and inflammaging. Immunosenescence is a phenomenon characterized by thymic involution, a reduced number of naive T lymphocytes, increased differentiated T, and a reduced number of B lymphocytes. Inflammaging is the condition for which the aging immune system tends to produce a greater amount of inflammatory mediators and therefore a more severe tissue damage [58].

### 3.2. Viral Factors

HRSV is one of the most contagious human pathogens, which is comparable to the measle virus [28]. Despite the low hRSV cytopathic effect, viral load appears to drive disease manifestations in humans with hRSV infection as well as host immunopathology [27,59,60].

It is important to note that people can get infected with hRSV multiple times during life. This is because, unlike other viruses, immunity to hRSV infection is not fully effective and is short-lived. This occurs because hRSV can modulate and escape the immune system in several ways. First, secretory protein G, acting as a decoy, mediates the virus evasion from antibody-mediated neutralization [20]. Furthermore, the two non-structural proteins NS1 and NS2 promote hRSV replication by inhibiting apoptosis during early infection. In addition, NS1 and NS2 negatively modulate dendritic cell maturation and T-cell responses by preventing the host cell from producing IFN-I [31,32,34]. The SH transmembrane protein also downregulates apoptosis [27,61].

### 3.3. RSV and SARS-CoV-2

The epidemiology of RSV was modified by the pandemic from SARS-CoV-2. In fact, the number of cases of infection with RSV has drastically collapsed in winter 2020, while with the reduction in the distancing measures, there has been an increase in RSV infections also off-season [62]. A recent narrative review took stock of the possible causes of these changes [63]. Summing up, the recrudescence of RSV infection cases is partly attributable to the so-called immune debt, which is a hypothesis supported by evidence of the reduction in RSV antibody titers [64,65,66]. No evidence from experimental models shows that SARS-CoV-2 or co-infection raises RSV cytopathogenicity in human bronchial epithelium cells [67]. Additionally, in cellular cultures, SARS-CoV-2 generated less effective anti-viral innate immune responses in comparison to RSV, as indicated by the interferon-stimulated genes’ expression [63,67]. Although there are some laboratory pieces of evidence of possible immune dysregulation following SARS-CoV-2 infection (e.g., a lower proportion of CD4 and CD8 naïve T lymphocytes and an increase in senescence markers in patients healed by COVID-19 [68,69]), clinical and epidemiological evidence are lacking [63]. Even the emergence of more transmissible and virulent RSV strains remains unproven at the moment [63].

## 4. Immune Response

### 4.1. Innate Immune Response

After airways’ penetration, hRSV infects epithelial ciliated cells and activates alveolar macrophages and dendritic cells, triggering an innate anti-viral response (Figure 3).

Specifically, innate immunity cells express several receptors, called pattern recognition receptors (PRRs), which recognize some pathogens’ molecular structures, called pathogen-associated molecular patterns (PAMPs) [17]. This ligand-receptor binding triggers the production of pro-inflammatory cytokines and interferon type I (IFN-I). The main known PRRs are Toll-like receptors (TLRs), RIG-I-like receptors (RLRs), and NOD-like receptors (NLRs). Among them, TLR2, TLR3, TLR4, TL6, TLR7, RIG-I and NOD2 can recognize RSV PAMPs [17]. The effect of hRSV and TLR-2/4/6 interaction is twofold: while it activates the NF-kB signaling pathway promoting the pro-inflammatory cytokines, it also increases TLR expression by epithelial cells and monocytes [71,72,73,74]. Lung plasmacytoid dendritic cells’ (pDC) expression of endosomal single-stranded RNA receptor (ssRNA) TLR-7 is also increased by hRSV infection. TLR7 activation results in the production of pro-inflammatory cytokines and IFN-I by pDCs [14]. The intracellular receptor RIG-I, binding the 5′ triphosphorylated viral RNA in the cytosol through a helicase domain, interacts with the mitochondrial anti-viral signaling protein (MAVS), inducing the production of pro-inflammatory cytokines and IFN-I through the NF-kB pathway. RIG-I recognizes RSV transcripts as ligands, and this interaction increases RIG-I expression in epithelial cells. The single-strand viral genome is also recognized by the NOD2 cytoplasmic receptor, whose activation promotes IFN-I production. In addition, NOD-like receptors/RSV-SH interaction induces inflammasome assembly in lung epithelium cells, which generates the cytokine IL-1b [17]. The innate response results in the production of pro-inflammatory cytokines, chemokines (IL-8, IL-4, IL-5, IL-6, IL-10, RANTES, TNF-α, IL-1α and IFN-α/λ), and adhesion molecules with the activation of natural killer cells (NK), granulocytes, monocytes, macrophages, dendritic cells, and T lymphocytes [75,76] (Figure 3).

#### 4.1.1. Dendritic Cells

Dendritic cells (DCs) are specialized antigen-presenting cells (APCs) that have a significant role in both innate immunity and adaptive immune response activation [17]. DCs detect viruses through innate receptors during viral infection and convert viral antigens into peptides, which are then presented as epitopes to T cells in a complex with MHC molecules [17]. Lung DCs are classified as follows: (I) DC1 or cDC1, generally considered the first subtype to present the antigenic epitope to CD8^+^ cytotoxic T lymphocytes; (II) cDC2 involved in communication with CD4^+^ T helper lymphocytes; (III) pDC responsible for the production of IFN-I [17,75,77]. Each type is widely present and distributed in the lungs. After interaction with pathogens, dendritic cells migrate to the lymph nodes through pulmonary lymphatic drainage to initiate the adaptive immune response [17,75,77] (Figure 3).

#### 4.1.2. Neutrophils

In patients affected by hRSV infection, neutrophils are the predominant cell type in the bronchial epithelium [33]. Recruited by IL-8, neutrophils migrate to respiratory epithelium expressing activation markers such as CD18 and ICAM-1 and causing damage tissue damage by the production of the neutrophil elastase protein (ELA-2) [33]. They also interact directly with RSV, internalizing the virus through the phagocytic process and allowing the virus replication within itself. In a study conducted over 22 infants younger than 52 weeks of age admitted to a pediatric ICU, neutrophil precursors peak in the blood followed by 2–3 days of viral load and symptoms peak (Figure 3) [70]. Infected infants’ neutrophil counts normalize upon ICU discharge [33,70]. Neutrophil infiltration is significantly present in the lung tissue of fatal cases of RSV LRTI [33].

#### 4.1.3. Eosinophils

There is evidence of eosinophil activity during the acute phase of RSV LRTI [33]. According to a preclinical study, neonatal mice with RSV infection experienced a worsening of allergic asthma when exposed to allergens. This was accompanied by an increase in the detection of eosinophils in the lungs [78]. However, there are currently no data demonstrating significant eosinophil recruitment to the respiratory tract in humans. During RSV bronchiolitis, substances such as leukotriene C4, eosinophil-derived neurotoxin (EDN), and eosinophil cationic protein (ECP) can become highly concentrated in the infants’ respiratory tract. Among those, leukotriene C4 and ECP can be detected in nasal fluid, while EDN and ECP can be found in lower airway secretions [79,80,81]. Levels of eosinophil chemoattractant CCL-5 (RANTES), ECP, and esotaxin show a correlation with eosinophil counts in the respiratory tract and escalate as the transition from acute illness to recovery takes place in RSV LRTI [33,82]. While these findings imply that eosinophils may play a part in resolving infections, it contradicts the idea that the Th2-biased response (see Section 4.2.1), which includes eosinophilia, might lead to more severe disease [83,84].

#### 4.1.4. Monocytes and Macrophages

During the initial stage of infection, monocytes are recruited to the airways’ epithelium where they are induced to produce type I IFN through the myeloid differentiation factor 88-mediated (Myd88) pathway, the TLR7, and the MAVS. Thus, monocytes contribute to both inflammation and immune-mediated damage as well as viral CD8-mediated clearance [85].

Alveolar macrophages infected by hRSV co-express RSV surface glycoprotein and potent immunomodulatory molecules such as HLA-DR, IL-1, and TNF- α, thus performing a dual function of antigen presentation and immunomodulation [86]. Under physiological conditions, there are two subpopulations of pulmonary macrophages: alveolar macrophages (AMs) at the surface of alveolar cells and interstitial macrophages (IMs) in the pulmonary stromata. During inflammatory states, AMs are identified as the macrophages residing in the airspace (RAMs), while IMs migrate toward the alveoli cavity and transform into the recruited airspace macrophages (RecAMs) [87]. Depending on the microenvironment, both RAMs and RecAMs can be polarized toward two different phenotypes: the pro-inflammatory phenotype M1 promotes inflammation and tissue damage, and the anti-inflammatory phenotype M2 contrasts inflammation and promotes tissue repair [87]. RSV can trigger AMs polarization through direct invasion, cytokines, and intercellular communication signaling. Concerning cytokines, IFN-γ promotes M1 polarization, and this mechanism’s magnitude is age-related. In fact, in adults, CD4 T cells express a high level of sialic acid-binding immunoglobulin agglutinin (Siglec-1) ligand CD43 that antagonize the monocytes signaling for CD4 T cells IFN-γ release [87,88]. Another age-related mechanism is the production of IFN-I by AMs during the inflammatory phase. Such production would be extremely minor in children, thereby reducing the capacity for viral clearance [87]. These two differences partly explain the different susceptibility of children and adults to developing RSV severe infections.

#### 4.1.5. NK Cells

NK lymphocytes in children with hRSV infection are increased, with an overexpressed NKG2D receptor observed at the pulmonary level, while the systemic NK cell count is reduced [33]. The NKG2D-ligand recognition promotes the activation of NK cells producing high amounts of IFN-γ, which is responsible for the lung immune lesion [89]. This is consistent with the evidence that NK cell depletion significantly reduces the infiltration of total inflammatory cells and the production of IFN-γ in bronchoalveolar lavage fluid [89].

### 4.2. Adaptive Immune Response

As explained above, the activation of the innate immune response through the PAMPs and PRRs interaction triggers the release of cytokines and other signals that stimulate antigen-presenting cells (APCs) like dendritic cells. Dendritic cells migrate into the lymph nodes and act as APCs, processing and presenting antigens to naive CD4 and CD8 T cells. This interaction initiated a targeted adaptive immune response involving T and B cells and consisting of humoral and cell-mediated immunity [77].

The role of humoral immunity in developing short-lived protective antibodies is well demonstrated; mediated cell immunity would play pivotal roles in both hRSV clearance and pathogenesis [90]. Alternatively, recent findings from preclinical studies indicate that CD8^+^ T cells and CD4^+^ T cells could proficiently and autonomously manage RSV replication in human lung tissue even in the absence of a specific antibody response against RSV [91]. An initial transient systemic T-cell lymphopenia occurs during RSV LRTI; the underlying mechanism seems to be virus-induced T-cells apoptosis [84]. During acute infection, there is an upregulation of Fas and TRAIL receptor expression on circulating CD4^+^ and CD8^+^ T cells together with an increase in the systemic soluble Fas ligand and caspase-1 concentrations [84]. These findings show an inverse correlation with age and a direct association with infection severity. T-cell lymphopenia is more evident in younger individuals, severe illness cases, and patients necessitating ventilatory support [33,84].

#### 4.2.1. CD4 Lymphocytes

CD4^+^ T helper lymphocytes play a leading role in hRSV immune response and pathogenesis with four possible phenotype polarizations: Th1, Th2, Th17 and Treg. The microenvironment plays a pivotal role in orchestrating the polarization of T-helper (Th) cells and shaping the nature of immune responses. Within the microenvironment, various factors such as cytokines, antigen presentation, and interactions with neighboring cells exert profound influences on the differentiation and functional characteristics of Th cells [92].

The Th1 phenotype constitutes the specific adaptive immune response to intracellular pathogens. It induces the production of pro-inflammatory mediators like IFN-γ, IL-1, IL-12, IL-2, IL-18, and TNF-α [33]. Among those, IL-12 induces the secretion of IFN-γ, which contributes to reducing viral replication and favoring the Th1 cells differentiation (Figure 3) [33]. A recent preclinical study was conducted with the aim of characterizing the cellular immune response to RSV with the goal of developing a targeted T-cell therapy for convenient administration to immunocompromised individuals [93]. The study revealed that nucleoprotein N and protein F were the most potent antigens in terms of immunogenicity, leading to the in vitro activation of T cells that specifically recognize the RSV virus (RSV-STs). These RSV-STs were a diverse mixture of CD4^+^ and CD8^+^ T cells exhibiting a memory phenotype, which is characterized by the presence of central (CD45RO^+^/CD62L^+^) and effector memory (CD45RO^+^/CD62L^−^) markers. The effectiveness of RSV-STs against the virus was demonstrated through several indicators. These included the increased expression of molecules associated with degranulation (CD107a), co-stimulation (4-1BB), and activation (CD69) as well as the production of effector cytokines (IFNγ and TNF-α) upon exposure to viral antigens. The expanded RSV-STs were skewed toward a Th1-polarized response, displaying polyfunctional attributes and the ability to selectively target and eliminate HLA-matched antigen-loaded cells [93].

Secondly, CD4^+^ lymphocytes polarization toward the Th2 phenotype promotes the secretion of pro-inflammatory cytokines like IL-4, IL-5, IL-6, IL-9, IL-10, and IL-13, resulting in IgE production, eosinophils, neutrophils, and monocytes recruitment and Th1 response inhibition [84]. This response is specific for parasitic infections and is known to be related to allergic disease predisposition [33]. In children with severe hRSV bronchiolitis, IL-18 (inducing Th2 response) and IL-4 concentration (induced by Th2) are higher than IL-12 (inducing Th2 response) and INF-γ (induced by Th1) consistently with a reduced number of circulating Th1 lymphocytes (CXCR3^+^) compared to Th2 (CCR4^+^) [83,94]. These current data suggest that an imbalance in type 1/type 2 cytokines with deficient Th1 and excess Th 2 responses are associated with developing severe hRSV infections [83].

Lymphocytes Th17 are a specialized subset of CD4^+^ T helper lymphocytes primarily associated with defense against extracellular bacteria and fungi, particularly those that target mucosal surfaces, such as the respiratory and gastrointestinal tracts. They also play a role in tissue inflammation and autoimmunity, and they are characterized by their production of interleukin-17 (IL-17) and related cytokines [95]. Another mechanism involved in both the development of severe infections and the predisposition to recurrent wheezing and asthma is represented by the Th17 response induced by RSV infection [96]. In fact, RSV infection would favor the IL-17 production and the development of a Th17 response both involved in the recruitment of neutrophils in lung tissue and the promotion of inflammation by cytokines production [97]. In addition, Th17 would interfere with CD8-mediated viral clearance [97,98]. Based on recent evidence, RSV infection asthma susceptibility could be related to Th17/Treg ratio augmentation rather than Th1/Th2 imbalance [98].

Lymphocyte T CD4 T-reg (Tregs) are a specific subset of CD4^+^ T cells that play a critical role in immune regulation and maintaining immune tolerance. Tregs help to control the immune system by suppressing the activation and function of other immune cells, including CD4^+^ and CD8^+^ T cells [99]. Tregs are characterized by the expression of specific cell surface markers such as CD25 and the transcription factor FOXP3. Levels of circulating FOXP3 mRNA and the counts of FOXP3^+^ CD4^+^ regulatory T cells, which encompass both suppressive resting Treg cells (CD45RA^+^ FOXP3lo) and suppressive activated Treg cells (CD45RA^+^ FOXP3hi), are diminished in infants who are hospitalized due to RSV bronchiolitis. This reduction persists for a minimum of three weeks after the acute infection. It remains uncertain whether this decrease could signify apoptosis or the migration of these cells to the lungs [33]. Thus, in mice, the depletion of Treg before the infection causes a delayed release of the virus and a consequent more severe clinical case [100].

#### 4.2.2. CD8 Lymphocytes

Throughout the infection, there is a more significant expansion of CD8^+^ T cells compared to CD4^+^ T cells. These CD8^+^ T cells display an effector phenotype marked by characteristics such as HLA-DR positivity, granzyme B presence, and CD38 expression [101,102] (Figure 3). In infants with severe RSV LRTI who require mechanical ventilation, the levels of systemic effector CD8^+^ T cells are low when symptoms and viral load are at their peak [70] (Figure 3). However, the CD8^+^ T-cell counts peak during the recovery phase, following the systemic neutrophil response. When infants are discharged from the ICU, circulating CD8^+^ T-cell counts experience a temporary increase, while neutrophil levels return to normal (Figure 3) [70]. Preclinical investigations indicate that depleting human CD8^+^ T cells in mice hampers though does not entirely eradicate the ability to control RSV infection. This implies that CD8^+^ T cells alone are not the exclusive agents responsible for virus clearance [91]. Likewise, findings from preclinical studies suggest that CD8^+^ T cells could also play a role in causing lung damage in humans after being exposed to RSV [91].

#### 4.2.3. Humoral Immune Response

Concerning the humoral response, hRSV infection is characterized by the increase in circulating B cells, both mature (CD19^+^ and CD5^+^) and precursors (CD19^+^ and CD10^+^), CD20^+^ B cells, and IgM^+^, IgA^+^ and IgG^+^ plasma cells [33,103]. T-independent mechanisms mainly carry out the stimulation of B lymphocytes and antibody production (Figure 3). In fact, during RSV infections, high levels of B cell stimulating factors, proliferation-inducing ligands (APRIL), and B cell activation factors (BAFF) have been found in the respiratory epithelium. The high concentrations of these factors are positively related to the levels of RSV IgA, IgG, and IgM in the lung [103]. The antibody response is mainly directed against viral surface glycoproteins such as proteins G and F [104]. The serum-neutralizing antibodies and the mucosal IgA and IgG toward the G glycoprotein are RSV group-specific, whereas the antibodies targeting the F glycoprotein exhibit cross-reactivity across different RSV groups [105]. Secretory IgA plays a fundamental role in protecting the upper respiratory tract both in the short-lived response following a primary infection and the long-lasting response in reinfection episodes [106]. The development of the IgA response appears to correlate with recovery both in experimental and natural infections [107,108]. A study of 61 healthy adult patients showed that mucosal IgA levels correlate with RSV-reinfection susceptibility unlike neutralizing serum Ig [108]. Furthermore, a recent study of two cohorts of adult patients, young (age 18–55) and elderly (age 60–75), found that after experimental infection, the levels of post-F and pre-F nasal IgA did not increase in older participants even though they showed strong serum IgG response, and this finding was associated with higher viral loads in elderly patients. It also demonstrated that serum IgG was linked with protection from reinfections in older adults but not in younger individuals [109]. During primary RSV infection in young infants, maternal RSV IgG antibodies could potentially inhibit the generation of IgA F-targeted antibody response at respiratory infection sites. These observations could be linked to the severity of the initial infection and the extended recovery period often witnessed in young infants affected by RSV infection [52,53]. Serum IgG ensures substantial but limited protection against RSV infections; IgG’s short life and viral immune evasion strategies explain the high incidence of reinfections following primary infection. Lastly, the IgE-mediated response against RSV-F and RSV-G seems to have a deleterious role: it has been found that in children with RSV-induced bronchiolitis or pneumonia, high levels of IgE in serum are associated with a prolongation of fever, worsening of symptoms and the appearance of rales [110].

#### 4.2.4. Immunological Memory

Regarding the duration of antibody protection, two studies evaluated the kinetics of antibody titer induced by natural hRSV re-infection in the adult population. The first study showed more than a fourfold drop in titer in 75% of subjects in one year [111]. This titer decline was significantly faster than the antibody decline rate in uninfected subjects, and it is still unclear if this rapid decline persists for longer than two years (study observation period) [111]. In the second study, patients exhibited three distinct antibody kinetics profiles related to RSV: uninfected, acutely infected, and recently infected, which were indicative of their RSV infection status at the enrollment [112]. Even if acutely infected and recently infected subjects showed similar mean neutralizing antibody titer peak responses, the recently infected group had a significant precipitous decrease in RSV antibodies in only 60 days. Among subjects with acute RSV infection, the levels of functional and binding RSV-specific antibodies remained consistent throughout the following 125 days except for a notable reduction in the competitive antibody concentration targeting the prefusion site Ø of the F-protein [112]. Later, the longevity of RSV-specific memory T cell responses to the F protein following natural RSV infection was measured in the same cohort of patients. Uninfected subjects showed stable memory T cell responses and polyfunctionality, while both the acutely and recently infected groups had reduced T cell polyfunctionality compared to the uninfected group at enrollment. In particular, the acutely infected group infection showed higher PD-1 (exhaustion T cells marker) expression in their T cells, especially at enrollment and even without stimulation. This suggests that their T cells may have been exhausted before the infection, making them more susceptible to RSV reinfection. Otherwise, for acutely infected subjects, the memory T cell response returned comparable levels to the uninfected group levels at the end of the season [113]. In this study, the authors postulated that the antibody production in recently infected patients might be orchestrated by short-lived circulating plasma blasts. These plasma blasts have the capacity to swiftly generate substantial antibody quantities after infection, as opposed to the long-lived plasma cells that usually inhabit the bone marrow and sustain elevated antibody levels over extended periods. All these observations would suggest the possibility that some subjects are predisposed to develop an immune response, both humoral and cell-mediated, not lasting over time [113]. Why this happens in some patients remains unclear for the moment, but the knowledge of this immunological profile could help in the identification of subjects at higher risk of reinfection. A further study carried out on 61 healthy adult patients showed a defect in mucosal memory mechanisms. As part of the humoral response to RSV, only IgG and no IgA RSV-specific memory B cells were found in peripheral blood after RSV infection, other than as found in influenza infections [108]. Given the proven correlation between mucosal IgA and RSV reinfection prevention, this could at least partly explain the recurrence of RSV reinfection over life [108].

Concerning cell-mediated response, RSV was shown to inhibit the development of memory CD8 T lymphocytes during acute infection via IL-21–STAT3 pathway impairment [114,115]. In fact, in a 2019 study, the STAT3 expression level in the nasal wash of infants diagnosed with bronchiolitis was significantly lower than in infants infected by other respiratory viruses. In the same study, in silico tests suggested that this mechanism could be mediated by protein G and IL21R interaction [115]. Moreover, a recent study was conducted on 2- and 3-year-old children to evaluate the memory T CD4 and CD8 cell responses to in vitro stimulation with RSV 3–12 by measuring transcription factors (T-bet, RORgt, GATA3) and cytokines (IFN-g, IL-2, TNF) expression. It was found that type-1 (IFN-g, IL-2) and type-17 (RORgt) markers expression was significantly lower in children infected with RSV during the first year of life than in children uninfected during infancy [116]. Concerning lung-resident memory T cells, a recent preclinical study in mice proved that lung-resident memory T cell differentiation is age-related and lowest in neonatal mice [117] but it might be increased with age, as well as other mechanisms of innate immunity, and this can result in lower viral load during reinfection [117]. This is in agreement with a further preclinical study showing that in mice defective for MAVS and Myd88, the expansion of tissue memory TCD8 lymphocytes is deficient and the susceptibility to RSV reinfections is greater [118].

## 5. RSV Prevention Strategies

In recent years, substantial advances have been made in developing RSV prevention strategies, such as mAbs and vaccines. Four products were finally approved, while several other candidates are in late-phase clinical development [119].

### 5.1. Monoclonal Antibodies

The introduction of passive immunization programs focused on safeguarding the first six months of life against RSV could lead to a substantial reduction in the burden of hRSV-related diseases [4]. The first preventive strategy developed against severe hRSV infections was polyclonal human intravenous immunoglobulin containing high concentrations of RSV protective antibodies (IVIG-RSV). For high-risk infants, this preventive strategy was linked to a 40% decrease in RSV-related hospitalizations, a 50% reduction in hospital stay duration, and a 60% reduction in the number of days requiring oxygen support [10]. However, production challenges, the task of finding high NT plasma donor populations, and the cost and difficulty of administering IVIG-RSV have limited the spread of such a preventive strategy. Research then focused on developing specific RSV mAb.

The first anti-RSV mAb approved was Palivizumab, which was licensed in June 1998 by the Food and Drug Administration to reduce severe lower respiratory tract infections caused by hRSV in children at increased risk of severe disease [10]. Palivizumab is a humanized mAb against RSV fusion (F) glycoprotein, and it is currently the most widely used prophylaxis for preventing RSV disease in infants. The medication is given through intramuscular injection on a monthly basis throughout the infant’s first RSV season with up to five doses administered at a rate of 15 mg/kg. This regimen aims to prevent severe RSV lower respiratory tract infections. In certain instances, children who have bronchopulmonary dysplasia or congenital heart disease might receive the drug for a second season as well [120].

The results of two meta-analyses on randomized clinical trials (RCTs) indicated that Palivizumab led to a notable reduction in RSV-related hospitalizations with a decrease of 51 to 55 hospitalizations per 1000 participants (baseline risk: 98–101 hospitalizations per 1000 participants) in comparison to the placebo group [120,121]. A second mAb, Nirsevimab, was approved in the EU on 3rd November 2022 to prevent hRSV lower respiratory tract disease in neonates and infants during their first hRSV season [122]. Nirsevimab is a long-acting recombinant neutralizing human IgG1ĸ mAb capable of binding to a highly conserved RSV fusion protein subunits, the F1 and F2 epitope. This binding blocks the fusion protein in the prefusion conformation, thus preventing the virus from entering the bronchial epithelial cells [12]. A triple amino acid substitution (YTE) in the Fc region can extend the serum half-life as long as it allows Nirsevimab administration as a single dose (50 mg for infants with body weight <5 kg, 100 mg for infants with body weight ≥5 kg) to cover the RSV season [122]. In the beginning, the effectiveness and safety of Nirsevimab were assessed in a randomized, placebo-controlled clinical trial involving preterm infants. The trial revealed that over the span of 150 days following administration, there was a substantial decrease in the incidence of RSV-associated medically attended lower respiratory tract infections (MALRTIs) and the rate of hospitalization due to RSV-associated MALRTI in the treated group compared to the control group with reductions of 70% and 78%, respectively [123]. Later, Nirsevimab was tested in late preterm and term infants in a randomized, placebo-controlled phase 3 trial. In this study, the efficacy of Nirsevimab in preventing RSV-associated MALRTIs was 74.5%, while the effectiveness in the reduction in hospitalization was 62.1% [12].

### 5.2. Vaccines

In May 2023, the US Food and Drug Administration approved two recombinant RSV prefusion F protein-based vaccines to prevent lower respiratory tract disease in individuals 60 years of age and older [124,125]. RSVPreF3 OA is an adjuvated RSV vaccine containing recombinant respiratory syncytial virus glycoprotein F stabilized in pre-fusion conformation (RSVPreF3) as the antigen component. In a recent RCT, it was demonstrated that a solitary administration of the vaccine displayed a satisfactory safety profile. The vaccine exhibited an efficacy of approximately 94.1% against severe RSV-related lower respiratory tract disease and 71.7% against RSV-related acute respiratory infection, regardless of the RSV subtype and the presence of underlying co-existing conditions [126]. The second authorized vaccine is a bivalent recombinant subunit vaccine featuring a stabilized prefusion F protein (RSVpreF). This vaccine includes balanced quantities of stabilized prefusion (preF) antigens derived from the two primary RSV subgroups (RSV A and RSV B). It demonstrated a 67% efficacy in lowering the likelihood of RSV-associated lower respiratory tract disease (LRTD) development in older adults who presented two or more symptoms. Additionally, it exhibited an 86% effectiveness in diminishing the risk of RSV-associated LRTD among individuals experiencing three or more symptoms [127].

Another strategy being studied is the vaccination of pregnant women to prevent RSV severe infections in newborns and infants. Recently, the results of a randomized, double-blind, controlled clinical trial of 3682 women and their children were published. In this study, the RSVpreF bivalent experimental vaccine containing stabilized preF glycoproteins from the two main circulating antigen subgroups (RSV A and RSV B) showed efficacy in preventing severe RSV LRTI by 81.8% (CI 99.5%, 40.6 to 96.3) within 90 days after birth and 69.4% (CI 97.58%, 44.3 to 84.1) within 180 days of birth [128].

## 6. Conclusions

RSV infection is one of the major challenges among infectious diseases with a significant impact on public health. It remains a leading cause of hospitalization among young children, immunocompromised people, and elders. The immune response to hRSV infection and its protective and pathogenetic role has been only partially clarified. Still, many dark points persist, such as the duration of humoral protection, the viral immune-escape mechanisms, the maternal antibody-induced immunodepression, and the role of IgE and Th2 response, so further investigations are needed. On the other hand, identifying the critical role played by neutrophils, Th1 response, and the production of antibodies against G and F proteins has allowed the development of promising prevention strategies.

## Figures and Tables

**Figure 1 pathogens-12-01118-f001:**
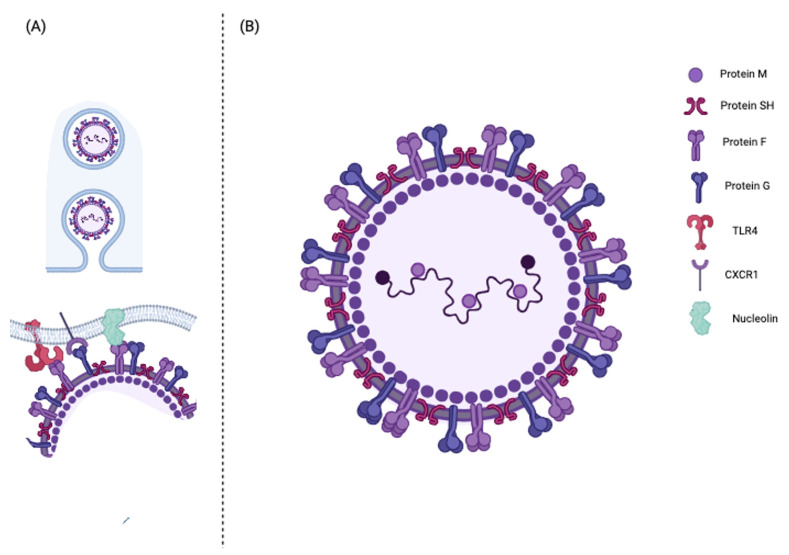
(**A**) Virus entry into the host cell: the adhesion to the cell membrane of the ciliate cells of the respiratory epithelium is mediated by the interaction between F and G proteins and some membrane receptors such as TLR4, CXCR1, and nucleolin. (**B**) HRSV molecular structure. Create in BioRender.com. (accessed on 14 August 2023).

**Figure 2 pathogens-12-01118-f002:**
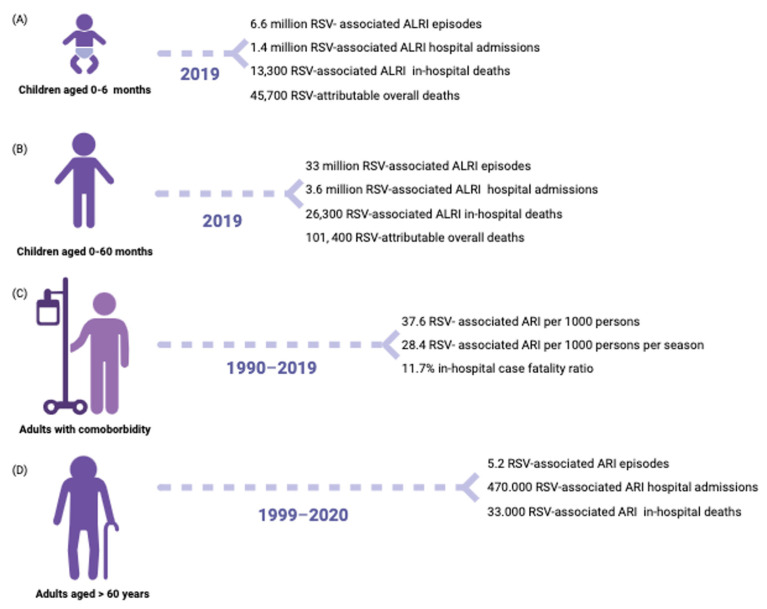
Global burden of RSV infection disease among children population from 0 to 60 months (**A**,**B**) [4]. Burden of RSV infection disease in high-income countries among adults with comorbidity (**C**), and elders (**D**) [8,9]. Create in BioRender.com. (accessed on 14 August 2023).

**Figure 3 pathogens-12-01118-f003:**
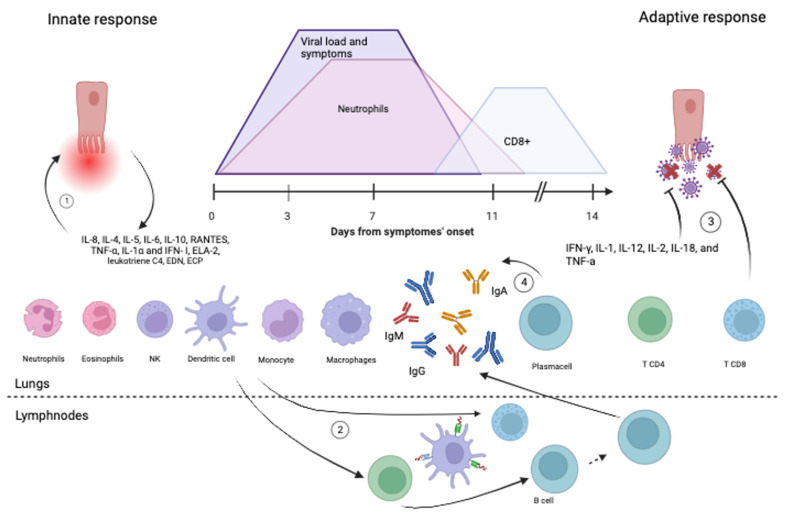
Innate and adaptive immune response to hRSV infection, inspired and modified from reference [21]. Graphic: The peak of the viral load and the severity of clinical manifestations is expected between the second and the fourth day after the onset of symptoms. A similar trend, with a 2–3 day delay, is typical of the blood neutrophils precursors. CD8 T cells peaked between the 11th and 14th days [70]. (**1**) Cellular infection triggers the release of early inflammatory mediators (e.g., IFNs, TNF-α, and chemokines), which recruit natural killer (NK) cells and polymorphonuclear leukocytes (PMNs) that have the ability to kill the infected cells, thus limiting viral replication but contribute to tissue damage. (**2**) Dendritic cells carry viral antigens to regional lymph nodes. Presentation of viral antigens to CD4^+^ T-lymphocytes occurs, and primed T-cells activate B-lymphocytes and CD8^+^ T-cells. They all migrate back to the infected epithelium with the further release of mediators and recruitment of additional inflammatory cells, including PMNs and mononuclear cells. (**3**) CD4-Th1 response and CD8 cytotoxic activity contribute to viral clearance. (**4**) Antibodies are produced by both T-cell-independent and T-cell-mediated B lymphocyte activation. Create in BioRender.com. (accessed on 14 August 2023).

**Table 1 pathogens-12-01118-t001:** HRSV virulence factors.

Virulence Factor	Protein Type	Mechanism
N	Capsid protein	Protecting viral RNA from nuclease, regulating viral RNA transcription and replication [14,15]
P	Polymerase cofactor phosphoprotein	Regulating viral RNA transcription and replication [14,15]
L	RNA dependent RNA polymerase	Mediating viral RNA transcription and replication [15,16]
MM2-1	Matrix protein	Regulating viral RNA transcription as elongation factor, mediating virions assembly [17,18]
M2-2	Matrix protein	Inhibiting viral RNA transcription, assuring balance between viral transcription and replication [17,18]
G	Membrane glycoprotein	Mediating adhesion to host cells’ membrane and antibody neutralization (as secretory form) [19,20,21]
F	Membrane glycoprotein	Mediating entry in the host cells [22,23]
SH	Small hydrophobic membrane protein	Increasing the membrane’s permeability in the host cells, inhibiting infected cells’ apoptosis [24,25,26,27,28]
M	Phosphorylated matrix protein	Mediating virion assembly, increasing virus replication trough host cell transcription inhibition and cell cycle arrest in G1 phase [29,30]
NS1, NS2	Non-structural proteins	Inhibiting IFNI/III production, dendritic cells activation, T-cell response and, infected cells’ apoptosis [31,32]

**Table 2 pathogens-12-01118-t002:** HRSV receptors expressed by airways epithelium ciliated cells and tipe-1 pneumocytes.

Receptor	Viral Protein Ligand	Functions
CXCR3	Protein G	Virus adhesion, Th2 response, IFN I production inhibition [36,37,38,39,40]
HSPGs	Protein G and F	Virus adhesion [36,37]
Nucleolin	Protein F	Virus internalization [23,42,43]
TLR4	Protein F	Endocytosis pathway activation [36,37,41]
ICAM-1	Protein F	Virus adhesion, neutrophils and eosinophils adhesion to the airway epithelium [36,37]
IGFR1	Protein F	Virus internalization [36,37]
EGFR	Protein F	Endocytosis pathway activation, epithelial cells fusion (syncytia) and mucus secretion [36,37]

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
