# Peer review of "Reshaping Our Knowledge: Advancements in Understanding the Immune Response to Human Respiratory Syncytial Virus"

_pathogens, 2023, doi:10.3390/pathogens12091118_

Round 1

Reviewer 1 Report

The authors have reviewed the literature of HRSV, RSV infection epidemiology, RSV immune responses, and strategies to mitigate severe RSV infections and how to reduce RSV related morbidity and mortality. The authors conclude that further efforts are needed to improve the understanding of the longevity of anti-RSV humoral responses, viral immune evasion mechanisms and the role of maternal anti-RSV IgG in the newborn.

Each part of the manuscript is well written and easy to understand. The two figures are informative. It is an interesting review that aim to cover the anti-RSV immunity and RSV induced immunopathology as well as how to decrease the impact of RSV-infection on public health. The manuscript is interesting but should be improved. My specific comments are listed below.

 Major points:

1.       Change the title to a statement rather than a question.

2.       Refer to figures in the text.

3.       Authors should avoid inexact phrasings such as lung cells.

4.       Many of the references are old: Of 95 references only 14 were published 2020 or later. Of the 14 more recent references 6 referred to newly approved anti-RSV antibodies, drugs, or vaccines and another 4 were review papers. This is surprising given the increase in knowledge about human respiratory tract viral infections made since the outbreak of the COVID-19 pandemic. Authors should update the references and discuss similarities and differences between RSV and SARS-CoV-2-infection, or at least mention why they don’t discuss SARS-CoV-2.

5.       Consider adding a table with RSV virulence factors

6.       Old references are also reflected in text where well-established knowledge such as M1/M2 macrophages and Th17 are missing when discussing anti-viral immune responses. Information of how innate responses govern adaptive responses in viral infection is missing.

7.       The authors should more clearly describe the study populations (usually hospitalized and in several cases ICU patients) of the referred papers and how the selection may affect the results. Also, the authors should more clearly indicate whether they refer to a systemic response versus a local response in the airways, and potential implications. What is known about IgA against RSV produced in locally in the airways?

8.       Is there any current knowledge about T cell exhaustion/senescence in RSV?

9.       Information about why elderly are more susceptible to severe RSV-infection is missing. It is only stated in line 37.

10.    The authors should also discuss the strategy to vaccinate pregnant women to protect newborn babies from RSV-infection.

11.    On line 183-185 authors write that young infants are more vulnerable to respiratory tract infections partly because of a lower magnitude and less effective immune responses. Can a weaker immune response be an advantage in RSV-infection?

 Minor points:

1.       Line 43. BPD, please spell out.

2.       Line 91. Rephrase “an antibody-type immune response”

3.       Line 130 paragraph 2.2. What receptor(s) is/are the most important for viral entry and what cells express this/these receptor(s)? Please clarify, table could help.

4.       In what cell types does RSV establish a productive infection?

5.       Paragraph 3.1 Consider dividing into three paragraphs: otherwise healthy young infants, children/people living with primary or secondary immunodeficiencies and add information about why elderly people are hit harder by RSV.

6.       Line 177-179. Please write more clearly about what genetic polymorphisms that are of importance. IL-4, IL-8 are not receptors!

7.       Line 235-238. Try to be more specific when describing the initiation of the innate response. Or state that the details are still not known.

8.       Why are macrophages and NK-cells found in the same paragraph? What about lymphocyte:monocyte ratio in RSV infection? Consider including monocytes in figure 2.

9.       Line 296. Consider to describe the role of CD4+ T helper cells before CD8 T cells.a

10.    Line 343. Usually, B cells are characterized as naïve, memory or switched memory. What do you mean by mature and precursors (circulating immature cells is an indicative of leukemia)?

11.    Line 370. Consider changing -0.20log2/month to something that is easier to comprehend, so the reader doesn’t need to open the calculator.

12.    Line 385. VRS? Also add 1-2 reference(s).

13.    Line 403. Type-17 needs to be explained. You should also mention type-2 responses here in this context.

Author Response

Reviewer 1

Author's response

Thank you for your thorough review of our article which gives us the opportunity to improve our work. We appreciate your valuable feedback and would like to address each of your points in detail (in red).

Major points:

1R. Change the title to a statement rather than a question.

1A. We have followed your suggestion and changed the title to a statement to enhance clarity.

2R. Refer to figures in the text.

2A. We have incorporated references to figures within the text for better context.

3R. Authors should avoid inexact phrasings such as lung cells.

3A. We have worked to eliminate any imprecise or vague terminology.

4R. Many of the references are old: Of 95 references only 14 were published 2020 or later. Of the 14 more recent references 6 referred to newly approved anti-RSV antibodies, drugs, or vaccines and another 4 were review papers. This is surprising given the increase in knowledge about human respiratory tract viral infections made since the outbreak of the COVID-19 pandemic. Authors should update the references and discuss similarities and differences between RSV and SARS-CoV-2-infection, or at least mention why they don’t discuss SARS-CoV-2.

4A. We have enriched the references with more recent works, as per your guidance. Additionally, we've included a brief paragraph discussing changes in RSV infection epidemiology observed during and after the COVID-19 pandemic. Concerning the similarities and differences between RSV and SARS-CoV-2, we've refrained from delving into this topic extensively, as our focus remains on the natural immune response against RSV in this narrative review.

5R. Consider adding a table with RSV virulence factors

5A. We have included a table (Tab.1) listing RSV virulence factors.

6R. Old references are also reflected in text where well-established knowledge such as M1/M2 macrophages and Th17 are missing when discussing anti-viral immune responses. Information of how innate responses govern adaptive responses in viral infection is missing. 

6A. We've integrated the macrophage paragraph with evidence regarding the two macrophages’ subpopulations and phenotypes (lines 331-348). Within the paragraph concerning T cell-mediated responses, we've also included evidence regarding the Th-17 response (lines 404-411).

7R. The authors should more clearly describe the study populations (usually hospitalized and in several cases ICU patients) of the referred papers and how the selection may affect the results. Also, the authors should more clearly indicate whether they refer to a systemic response versus a local response in the airways, and potential implications. What is known about IgA against RSV produced in locally in the airways?

7A. We've strived to clarify the study populations mentioned. Regarding mucosal IgA, we've included references to studies highlighting the correlation between secretory IgA development, reduced susceptibility to reinfection, and its deficiency in the elderly population.

8R.       Is there any current knowledge about T cell exhaustion/senescence in RSV?

8A.  Based on our research, we did not uncover substantial evidence regarding T cell senescence in RSV infection. The only reference in this context is mentioned in lines 484-489. This clinical study demonstrates an increase in T cell senescence markers in adults infected with RSV compared to non-infected individuals, which tends to regress during convalescence.

9R.       Information about why elderly are more susceptible to severe RSV-infection is missing. It is only stated in line 37.

9A. We've provided reasoning for the increased susceptibility of the elderly to RSV infection in lines 219-228.

10R.    The authors should also discuss the strategy to vaccinate pregnant women to protect newborn babies from RSV-infection.

10A. We've expanded the text to discuss the preventive strategy of vaccinating pregnant women to protect neonates against RSV infection in lines 586-592.

11R.  On line 183-185 authors write that young infants are more vulnerable to respiratory tract infections partly because of a lower magnitude and less effective immune responses. Can a weaker immune response be an advantage in RSV-infection? 

11A. While theoretically, a weaker immune response might be advantageous in minimizing tissue damage during acute infections, we have acknowledged that there is no evidence to support this notion in the context of RSV. In fact, populations with less effective immune systems, such as immunocompromised individuals, are at a higher risk of severe RSV infections.

Minor points:

  1. Line 43. BPD, please spell out. We have spelled out "BPD" as bronchopulmonary dysplasia in line with your request.
  2. Line 91. Rephrase “an antibody-type immune response”. The expression has been rephrased to "targeted antibody response" in line 91.
  3. Line 130 paragraph 2.2. What receptor(s) is/are the most important for viral entry and what cells express this/these receptor(s)? Please clarify, table could help. We have provided more specific information about the receptors involved in viral entry and introduced a table (Tab.2) to enhance clarity on this aspect.
  4. In what cell types does RSV establish a productive infection? We've specified that the cells in which RSV establishes a productive infection are ciliated epithelial cells of the bronchial epithelium and type I pneumocytes.
  5. Paragraph 3.1 Consider dividing into three paragraphs: otherwise healthy young infants, children/people living with primary or secondary immunodeficiencies and add information about why elderly people are hit harder by RSV. As per your suggestion, we have divided the paragraph on host factors into three sections. Additionally, we have elaborated on the susceptibility of elderly individuals to RSV infection in response to your feedback.
  6. Line 177-179. Please write more clearly about what genetic polymorphisms that are of importance. IL-4, IL-8 are not receptors! We've provided clearer details about the genetic polymorphisms associated with susceptibility to RSV infection.
  7. Line 235-238. Try to be more specific when describing the initiation of the innate response. Or state that the details are still not known. We've introduced a clearer introductory sentence regarding the initiation of the innate immune response and its mechanisms.
  8. Why are macrophages and NK-cells found in the same paragraph? What about lymphocyte:monocyte ratio in RSV infection? Consider including monocytes in figure 2. We have separated the discussion of macrophages and NK cells into separate paragraphs for better clarity. We've also included monocytes in the figure (formerly Figure 2, now Figure 3). Regarding the lymphocyte-monocyte ratio, we couldn't find specific literature evidence related to RSV.
  9. Line 296. Consider to describe the role of CD4+ T helper cells before CD8 T cells. Following your advice, we've reordered the sub-paragraphs to first describe the CD4+ T helper cell response before addressing CD8+ T cells.
  10. Line 343. Usually, B cells are characterized as naïve, memory or switched memory. What do you mean by mature and precursors (circulating immature cells is an indicative of leukemia)? The distinction between mature B cells and precursors is based on the markers CD5 for mature B cells and CD10 for precursors, as noted in the referenced study (ref. 98). It's not indicative of activation/commitment status.
  11. Line 370. Consider changing -0.20log2/month to something that is easier to comprehend, so the reader doesn’t need to open the calculator. We've made the phrasing in line 370 more user-friendly to eliminate the need for a calculator.
  12. Line 385. VRS? Also add 1-2 reference(s). We've corrected the typo "VRS" to "RSV" and enriched the paragraph with the results of two additional studies (ref. 112-113).
  13. Line 403. Type-17 needs to be explained. You should also mention type-2 responses here in this context. As mentioned earlier, we have included the discussion of the Th17 response in lines 404-411 and introduced a reference to the Th2 response in lines 514-515, elaborated further in section 4.2.1.

We are grateful for your guidance and have made the necessary revisions to enhance the quality and depth of our article. Your feedback has been helpful in refining our manuscript.

The authors

Reviewer 2 Report

The authors provide a brief, easy-to-read, review of what is known about the immune response to human Respiratory Syncytial Virus.  There only a few minor items for consideration.

1) Abstract, change "...characterize the immune response" to "summarize what is known about the immune response".  

2) spacing before references varies throughout the manuscript, in some instances there is no space between the last word and the reference, in other instances, there is a space.

3) An additional figure summarizing the epidemiology would be helpful.  Suggest age groups, hospitilization rates, and risk factors.

4) the figures should be closer to the relevant section of the text.  The "standard" schematic of the virus is at the end of the paper, several pages after it is described.  Figure 2 would benefit from a higher level of detail, perhaps include a timeline of the events such as CD8 peak, etc.

5) page 2 line 67.  Change "RNA polymerase dependent RNA enzyme L protein" to "of the L protein, the RNA dependent (or directed) RNA polymerase" 

The manuscript is well written and easy to follow.

Author Response

Thank you for your review of our article on the immune response to human Respiratory Syncytial Virus. We greatly appreciate your feedback and have addressed the points you've raised:

1) Abstract, change "...characterize the immune response" to "summarize what is known about the immune response".  We have amended the sentence in the abstract as suggested, as the proposed expression is more accurate and suitable.

2) spacing before references varies throughout the manuscript, in some instances there is no space between the last word and the reference, in other instances, there is a space. We have made the necessary adjustments to ensure consistent spacing before references throughout the manuscript.

3) An additional figure summarizing the epidemiology would be helpful.  Suggest age groups, hospitilization rates, and risk factors. An additional figure (Fig.2) has been included to provide a summary of recent global epidemiological data on RSV infection in children, adults with comorbidities, and the elderly.

4) the figures should be closer to the relevant section of the text.  The "standard" schematic of the virus is at the end of the paper, several pages after it is described.  Figure 2 would benefit from a higher level of detail, perhaps include a timeline of the events such as CD8 peak, etc. We have integrated references to the figures within the text, and while the final formatting will be managed by the publisher, we concur that it's beneficial for figures to be located near the relevant sections. We have enhanced the former Figure 2, now Figure 3, with additional details.

5) page 2 line 67.  Change "RNA polymerase dependent RNA enzyme L protein" to "of the L protein, the RNA dependent (or directed) RNA polymerase" . We have revised the expression as recommended to improve precision and clarity.

Your input has been significantly valuable in enhancing the clarity and quality of our article. We appreciate your review and have incorporated your suggestions to improve the manuscript.

The authors

Reviewer 3 Report

Human respiratory syncytial virus (hRSV) infection is one of the major viral diseases in human, with a significant impact on public health. The work by Attaianese et al. reviewed the research progress on the innate and adaptive immune response induced by hRSV infection. This work is a contribution and of interest to the community, the framework of this review manuscript is complete and clear. However, there were several minor questions should be addressed.

1. The annotations of Figure 1 and Figure 2 are missing in the main text. And the Created in biorender.com in the figure should be moved to the legends.

2. Line 175: There's an extra blank.

3. Please try to cite the latest research papers in the manuscript. 

No.

Author Response

Thank you for your review of our manuscript. We appreciate your feedback and have addressed the points you've raised (in red):

  1. The annotationsof Figure 1 and Figure 2 are missing in the main text. And the “Created in biorender.com” in the figure should be moved to the legends.We have included references to Figures 1 and 2 within the main text and moved the attribution "Created in biorender.com" to the figure legends.
  2. Line 175: There's an extra blank. The extra blank space in line 175 has been removed.
  3. Please try to cite the latest research papers in the manuscript. We have tried to incorporate more recent research papers into the manuscript to enhance its currency.

Your input has been very valuable in refining our manuscript. We thank you for your review and have incorporated your suggestions to improve the clarity and quality of the work.

The Authors

Round 2

Reviewer 1 Report

The authors have addressed my main concerns about the manuscript. I still have a couple of suggestions:

Line 126 It is not clear what “ones” refer to.

Line 127-129

Consider to move these sentences to a section about RSV immune response, for example as a add on to lines 237-238.

The non-structural ones NS1 and NS2 prevent the host cell from producing type I

Table 1. “Dendritic cells maturation” doesn’t read well.

Table 1 and table 2 – add references?

Line 151-152. Is the grammar correct?

Line 171 “sto” – typo?

Line 179. Is the grammar correct?

Line 227. Inflammatory instead of inflammation?

Line 243 RSV should replace SARS-CoV-2?

Line 305 acme – typo?

Line 322 Th2 is here mentioned for the first time, but the concept of different T helper subsets is not explained until the section about CD4 T cells. Please consider some editing.

Line 357-359. This sentence can be improved for better understanding.

Line 373-374 Here only two different Th-responses are mentioned Th1/Th2. Add info about Th17 and Tregs which are introduced later in the manuscript. Also a few words about the importance of the microenvironment for Th-polarization could be included.

Line 466 High levels of RSV-specific IgE?

Line 500-505 Edit to improve understanding.

Line 510 Choose another word than essays.

Line 511-515 Edit to improve understanding.

Line 515-522 The text is not linked to the rest of the section, maybe it is better to delete it? It is also very hard to follow what you want to say.

Line 591-592 What is IC?

The authors should carefully review the english language of the manuscript. There are many typos and some sections that may be difficult to understand.
